# Which Neural Net Architectures Give Rise to Exploding and Vanishing Gradients?

**Boris Hanin**
Department of Mathematics
Texas A& M University
College Station, TX, USA
bhanin@math.tamu.edu

## Abstract

We give a rigorous analysis of the statistical behavior of gradients in a randomly initialized fully connected network $\mathcal{N}$ with ReLU activations. Our results show that the empirical variance of the squares of the entries in the input-output Jacobian of $\mathcal{N}$ is exponential in a simple architecture-dependent constant $\beta$, given by the sum of the reciprocals of the hidden layer widths. When $\beta$ is large, the gradients computed by $\mathcal{N}$ at initialization vary wildly. Our approach complements the mean field theory analysis of random networks. From this point of view, we rigorously compute finite width corrections to the statistics of gradients at the edge of chaos.

## 1 Introduction

A fundamental obstacle in training deep neural nets using gradient based optimization is the exploding and vanishing gradient problem (EVGP), which has attracted much attention (e.g. [BSF94, HBF+01, MM15, XXP17, PSG17, PSG18]) after first being studied by Hochreiter [Hoc91]. The EVGP occurs when the derivative of the loss in the SGD update

$$W \quad \longleftarrow \quad W \ - \ \lambda \, \frac{\partial \mathcal{L}}{\partial W}, \tag{1}$$

is very large for some trainable parameters $W$ and very small for others:

$$\left| \frac{\partial \mathcal{L}}{\partial W} \right| \quad \approx \quad 0 \ \text{or} \ \infty.$$

This makes the increment in (1) either too small to be meaningful or too large to be precise. In practice, a number of ways of overcoming the EVGP have been proposed (see e.g. [Sch]). Let us mention three general approaches: (i) using architectures such as LSTMs [HS97], highway networks [SGS15], or ResNets [HZRS16] that are designed specifically to control gradients; (ii) precisely initializing weights (e.g. i.i.d. with properly chosen variances [MM15, HZRS15] or using orthogonal weight matrices [ASB16, HSL16]); (iii) choosing non-linearities that that tend to compute numerically stable gradients or activations at initialization [KUMH17].

A number of articles (e.g. [PLR+16, RPK+17, PSG17, PSG18]) use mean field theory to show that even vanilla fully connected architectures can avoid the EVGP in the limit of *infinitely wide* hidden layers. In this article, we continue this line of investigation. We focus specifically on fully connected ReLU nets, and give a rigorous answer to the question of which combinations of depths $d$ and hidden layer widths $n_j$ give ReLU nets that suffer from the EVGP at initialization. In particular, we avoid approach (iii) to the EVGP by setting once and for all the activations in $\mathcal{N}$ to be ReLU and that we study approach (ii) in the limited sense that we consider only initializations in which weights and biases are independent (and properly scaled as in Definition 1) but do not investigate

other initialization strategies. Instead, we focus on rigorously understanding the effects of finite depth and width on gradients in randomly initialized networks. The main contributions of this work are:

1. **We derive new exact formulas for the joint even moments of the entries of the input-output Jacobian in a fully connected $\mathrm{ReLU}$ net with random weights and biases.** These formulas hold at finite depth and width (see Theorem 3).

2. **We prove that the empirical variance of gradients in a fully connected $\mathrm{ReLU}$ net is exponential in the sum of the reciprocals of the hidden layer widths.** This suggests that when this sum of reciprocals is too large, early training dynamics are very slow and it may take many epochs to achieve better-than-chance performance (see Figure 1).

3. We prove that, so long as weights and biases are initialized independently with the correct variance scaling (see Definition 1), **whether the EVGP occurs** (in the precise sense explained in §3) **in fully connected $\mathrm{ReLU}$ nets is a function only of the architecture and not the distributions from which the weights and biases are drawn**.

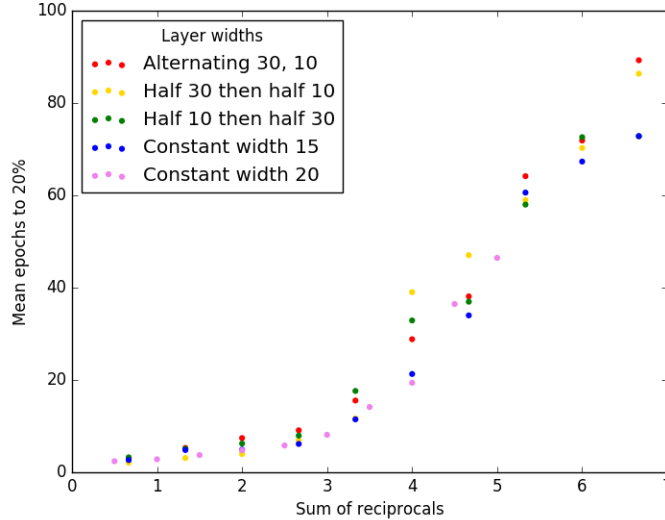

Figure 1: Comparison of early training dynamics on vectorized MNIST for fully connected $\mathrm{ReLU}$ nets with various architectures. Plot shows the mean number of epochs (over 100 independent training runs) that a given architecture takes to reach 20% accuracy as a function of the sum of reciprocals of hidden layer widths. (Figure reprinted with permission from [HR18] with caption modified).

## 1.1 Practical Implications

The second of the listed contributions has several concrete consequences for architecture selection and for understanding initial training dynamics in $\mathrm{ReLU}$ nets. Specifically, our main results, Theorems 1-3, prove that the EVGP will occur in a $\mathrm{ReLU}$ net $\mathcal{N}$ (in either the annealed or the quenched sense described in §3) if and only if a single scalar parameter, the sum

$$\beta = \sum_{j=1}^{d-1} \frac{1}{n_j}$$

of reciprocals of the hidden layer widths of $\mathcal{N}$, is large. Here $n_j$ denotes the width of the $j^{th}$ hidden layer, and we prove in Theorem 1 that the variance of entries in the input-output Jacobian of $\mathcal{N}$

is exponential in $\beta$. Implications for architecture selection then follow from special cases of the power-mean inequality:

$$\left( \frac{1}{d-1} \sum_{j=1}^{d-1} \frac{1}{n_j} \right)^{-1} \quad \leq \quad \frac{1}{d-1} \sum_{j=1}^{d-1} n_j \quad \leq \quad \left( \frac{1}{d-1} \sum_{j=1}^{d-1} n_j^2 \right)^{1/2}, \tag{2}$$

in which equality is achieved if and only if $n_j$ are all equal. We interpret the leftmost inequality as follows. Fix $d$ and a total budget $\sum_j n_j$ of hidden layer neurons. Theorems 1 and 2 say that to avoid the EVGP in both the quenched and annealed senses, one should minimize $\beta$ and hence make the leftmost expression in (2) as large as possible. This occurs precisely when $n_j$ are all equal. Fix instead $d$ and a budget of trainable parameters, $\sum_j n_j(n_{j-1} + 1)$, which is close to $\sum_j n_j^2$ if the $n_j$'s don't fluctuate too much. Again using (2), we find that from the point of view of avoiding the EVGP, it is advantageous to take the $n_j$'s to be equal.

In short, our theoretical results (Theorems 1 and 2) show that if $\beta$ is large then, at initialization, $\mathcal{N}$ will compute gradients that fluctuate wildly, intuitively leading to slow initial training dynamics. This heuristic is corroborated by an experiment from [HR18] about the start of training on MNIST for fully connected neural nets with varying depths and hidden layer widths (the parameter $\beta$ appeared in [HR18] in a different context). Figure 1 shows that $\beta$ is a good summary statistic for predicting how quickly deep networks will start to train.

We conclude the introduction by mentioning what we see as the principal weaknesses of the present work. First, our analysis holds only for ReLU activations and assumes that all non-zero weights are independent and zero centered. Therefore, our conclusions do not directly carry over to convolutional, residual, and recurrent networks. Second, our results yield information about the fluctuations of the entries $Z_{p,q}$ of the input-output Jacobian $J_{\mathcal{N}}$ at any fixed input to $\mathcal{N}$. It would be interesting to have information about the joint distribution of the $Z_{p,q}$'s with inputs ranging over an entire dataset. Third, our techniques do not directly extend to initializations such as orthogonal weight matrices. We hope to address these issues in the future and, specifically, believe that the qualitative results of this article will generalize to convolutional networks in which the number of channels grows with the layer number.

## 2   Relation to Prior Work

To provide some context for our results, we contrast both our approach and contributions with the recent work [PSG17, PSG18]. These articles consider two senses in which a fully connected neural net $\mathcal{N}$ with random weights and biases can avoid the EVGP. The first is that the average singular value of the input-output Jacobian $J_{\mathcal{N}}$ remains approximately 1, while the second, termed dynamical isometry, requires that all the singular values of $J_{\mathcal{N}}$ are approximately 1. The authors of [PSG17, PSG18] study the full distribution of the singular values of the Jacobian $J_{\mathcal{N}}$ first in the infinite width limit $n \to \infty$ and then in the infinite depth limit $d \to \infty$.

Let us emphasize two particularly attractive features of [PSG17, PSG18]. First, neither the initialization nor the non-linearity in the neural nets $\mathcal{N}$ is assumed to be fixed, allowing the authors to consider solutions of types (ii) and (iii) above to the EVGP. The techniques used in these articles are also rather general, and point to the emergence of universality classes for singular values of the Jacobian of deep neural nets at initialization. Second, the results in these articles access the full distribution of singular values for the Jacobian $J_{\mathcal{N}}$, providing significantly more refined information than simply controlling the mean singular value.

The neural nets considered in [PSG17, PSG18] are essentially assumed to be infinitely wide, however. This raises the question of whether there is any finite width at which the behavior of a randomly initialized network will resemble the infinite width regime, and moreover, if such a width exists, how wide is wide enough? In this work we give rigorous answers to such questions by quantifying finite width effects, leaving aside questions about both different choices of non-linearity and about good initializations that go beyond independent weights.

Instead of taking the singular value definition of the EVGP as in [PSG17, PSG18], we propose two non-spectral formulations of the EVGP, which we term annealed and quenched. Their precise definitions are given in §3.2 and §3.3, and we provide in §3.1 a discussion of the relation between the different senses in which the EVGP can occur.

Theorem 1 below implies, in the infinite width limit, that all ReLU nets avoid the EVGP in both the quenched and annealed sense. Hence, our definition of the EVGP (see §3.2 and §3.3) is weaker than the dynamical isometry condition from [PSG17, PSG18]. But, as explained in §3.1, it is stronger the condition that the average singular value equal $1$. Both the quenched and annealed versions of the EVGP concern the fluctuations of the partial derivatives

$$Z_{p,q} := \frac{\partial \left( f_{\mathcal{N}} \right)_q}{\partial \operatorname{Act}_p^{(0)}} \tag{3}$$

of the $q^{th}$ component of the function $f_{\mathcal{N}}$ computed by $\mathcal{N}$ with respect to the $p^{th}$ component of its input ($\operatorname{Act}^{(0)}$ is an input vector - see (10)). The stronger, quenched version of the EVGP concerns the *empirical variance* of the squares of all the different $Z_{p,q}$ :

$$\widehat{\operatorname{Var}}\left[ Z^2 \right] := \frac{1}{M} \sum_{m=1}^{M} Z_{p_m, q_m}^4 - \left( \frac{1}{M} \sum_{m=1}^{M} Z_{p_m, q_m}^2 \right)^2, \qquad M = n_0 n_d. \tag{4}$$

Here, $n_0$ is the input dimension to $\mathcal{N}$, $n_d$ is the output dimension, and the index $m$ runs over all $n_0 n_d$ possible input-output neuron pairs $(p_m, q_m)$. Intuitively, since we will show in Theorem 1 that

$$\mathbb{E}\left[ Z_{p,q}^2 \right] = \Theta(1),$$

independently of the depth, having a large mean for $\widehat{\operatorname{Var}}\left[ Z^2 \right]$ means that for a typical realization of the weights and biases in $\mathcal{N}$, the derivatives of $f_{\mathcal{N}}$ with respect to different trainable parameters will vary over several orders of magnitude, leading to inefficient SGD updates (1) for any fixed learning rate $\lambda$ (see §3.1 - §3.3).

To avoid the EVGP (in the annealed or quenched senses described below) in deep feed-forward networks with ReLU activations, our results advise letting the widths of hidden layers grow as a function of the depth. In fact, as the width of a given hidden layer tends to infinity, the input to the next hidden layer can viewed as a Gaussian process and can be understood using mean field theory (in which case one first considers the infinite width limit and only then the infinite depth limit). This point of view was taken in several interesting papers (e.g. [PLR$^+$16, RPK$^+$17, PSG17, PSG18] and references therein), which analyze the dynamics of signal propagation through such deep nets. In their notation, the fan-in normalization (condition (ii) in Definition 1) guarantees that we've initialized our neural nets at the edge of chaos (see e.g. around (7) in [PLR$^+$16] and (5) in [RPK$^+$17]). Indeed, writing $\mu^{(j)}$ for the weight distribution at layer $j$ and using our normalization $\operatorname{Var}[\mu^{(j)}] = 2/n_{j-1}$, the order parameter $\chi_1$ from [PLR$^+$16, RPK$^+$17] becomes

$$\chi_1 = n_{j-1} \cdot \operatorname{Var}[\mu^{(j)}] \int_{\mathbb{R}} e^{-z^2/2} \left( \phi'(\sqrt{q^*}z) \right)^2 \frac{dz}{\sqrt{2\pi}} = 1,$$

since $\phi = \operatorname{ReLU}$, making $\phi'(z)$ the indicator function $\mathbf{1}_{[0,\infty)}(z)$ and the value of $\phi'(\sqrt{q^*}z)$ independent of the asymptotic length $q^*$ for activations. The condition $\chi_1 = 1$ defines the edge of chaos regime. This gives a heuristic explanation for why the nets considered in the present article cannot have just one of vanishing and exploding gradients. It also allows us to interpret our results as a rigorous computation for ReLU nets of the $1/n_j$ corrections at the edge of chaos.

In addition to the mean field theory papers, we mention the article [SPSD17]. It does not deal directly with gradients, but it does treat the finite width corrections to the statistical distribution of pre-activations in a feed-forward network with Gaussian initialized weights and biases. A nice aspect of this work is that the results give the joint distribution not only over all the neurons but also over any number of inputs to the network. In a similar vein, we bring to the reader's attention [BFL$^+$17], which gives interesting heuristic computations about the structure of correlations between gradients corresponding to different inputs in both fully connected and residual ReLU nets.

## 3 Defining the EVGP for Feed-Forward Networks

We now explain in exactly what sense we study the EVGP and contrast our definition, which depends on the behavior of the *entries* of the input-output Jacobian $J_{\mathcal{N}}$, with the more usual definition, which depends on the behavior of its singular values (see §3.1). To do this, consider a feed-forward fully

connected depth $d$ network $\mathcal{N}$ with hidden layer widths $n_0, \ldots, n_d$, and fix an input $\mathrm{Act}^{(0)} \in \mathbb{R}^{n_0}$. We denote by $\mathrm{Act}^{(j)}$ the corresponding vector of activations at layer $j$ (see (10)). The exploding and vanishing gradient problem can be roughly stated as follows:

$$\text{Exploding/Vanishing Gradients} \quad \longleftrightarrow \quad Z_{p,q} \text{ has large fluctuations}, \tag{5}$$

where $Z_{p,q}$ the entries of the Jacobian $J_{\mathcal{N}}$ (see (3)). A common way to formalize this statement is to interpret "$Z_{p,q}$ has large fluctuations" to mean that the Jacobian $J_{\mathcal{N}}$ of the function computed by $\mathcal{N}$ has both very large and very small singular values [BSF94, HBF+01, PSG17]. We give in §3.1 a brief account of the reasoning behind this formulation of the EVGP and explain why is also natural to define the EVGP via the moments of $Z_{p,q}$. Then, in §3.2 and §3.3, we define two precise senses, which we call annealed and quenched, in which that EVGP can occur, phrased directly in terms of the joint moments of $Z_{p,q}$.

## 3.1 Spectral vs. Entrywise Definitions of the EVGP

Let us recall the rationale behind using the spectral theory of $J_{\mathcal{N}}$ to define the EVGP. The gradient in (1) of the loss with respect to, say, a weight $W^{(j)}_{\alpha,\beta}$ connecting neuron $\alpha$ in layer $j-1$ to neuron $\beta$ in layer $j$ is

$$\partial \mathcal{L} / \partial W^{(j)}_{\alpha,\beta} = \langle \nabla_{\mathrm{Act}^{(d)}} \mathcal{L}, J_{\mathcal{N},\beta}(j \to d) \rangle \, \mathrm{Act}^{(j-1)}_{\alpha} \, \phi'(\mathrm{Act}^{(j)}_{\beta}), \tag{6}$$

where $\phi'(\mathrm{Act}^{(j)}_{\beta})$ is the derivative of the non-linearity, the derivative of the loss $\mathcal{L}$ with respect to the output $\mathrm{Act}^{(d)}$ of $\mathcal{N}$ is

$$\nabla_{\mathrm{Act}^{(d)}} \mathcal{L} = \left( \partial \mathcal{L} / \partial \mathrm{Act}^{(d)}_q, \, q = 1, \ldots, n_d \right),$$

and we've denoted the $\beta^{th}$ row in the layer $j$ to output Jacobian $J_{\mathcal{N}}(j \to d)$ by

$$J_{\mathcal{N},\beta}(j \to d) = \left( \partial \mathrm{Act}^{(d)}_q / \partial \mathrm{Act}^{(j)}_{\beta}, \, q = 1, \ldots, n_d \right).$$

Since $J_{\mathcal{N}}(j \to d)$ is the product of $d - j$ layer-to-layer Jacobians, its inner product with $\nabla_{\mathrm{Act}^{(d)}} \mathcal{L}$ is usually the term considered responsible for the EVGP. The *worst case* distortion it can achieve on the vector $\nabla_{\mathrm{Act}^{(d)}} \mathcal{L}$ is captured precisely by its condition number, the ratio of its largest and smallest singular values.

However, unlike the case of recurrent networks in which $J_{\mathcal{N}}(j \to d)$ is $(d-j)-$fold product of a fixed matrix, when the hidden layer widths grow with the depth $d$, the dimensions of the layer $j$ to layer $j'$ Jacobians $J_{\mathcal{N}}(j \to j')$ are not fixed and it is not clear to what extent the vector $\nabla_{\mathrm{Act}^{(d)}} \mathcal{L}$ will actually be stretched or compressed by the worst case bounds coming from estimates on the condition number of $J_{\mathcal{N}}(j \to d)$.

Moreover, on a practical level, the EVGP is about the numerical stability of the increments of the SGD updates (1) over all weights (and biases) in the network, which is directly captured by the joint distribution of the random variables

$$\{ |\partial \mathcal{L} / \partial W^{(j)}_{\alpha,\beta}|^2, \, j = 1, \ldots, n_d, \, \alpha = 1, \ldots, n_{j-1}, \, \beta = 1, \ldots, n_j \}.$$

Due to the relation (6), two terms influence the moments of $|\partial \mathcal{L} / \partial W^{(j)}_{\alpha,\beta}|^2$: one coming from the *activations* at layer $j-1$ and the other from the *entries* of $J_{\mathcal{N}}(j \to d)$. We focus in this article on the second term and hence interpret the fluctuations of the entries of $J_{\mathcal{N}}(j \to d)$ as a measure of the EVGP.

To conclude, we recall a simple relationship between the moments of the entries of the input-output Jacobian $J_{\mathcal{N}}$ and the distribution of its singular values, which can be used to directly compare spectral and entrywise definitions of the EVGP. Suppose for instance one is interested in the average singular value of $J_{\mathcal{N}}$ (as in [PSG17, PSG18]). The sum of the singular values of $J_{\mathcal{N}}$ is given by

$$\mathrm{tr}(J_{\mathcal{N}}^T J_{\mathcal{N}}) = \sum_{j=1}^{n_0} \langle J_{\mathcal{N}}^T J_{\mathcal{N}} u_j, u_j \rangle = \sum_{j=1}^{n_0} \| J_{\mathcal{N}} u_j \|^2,$$

where $\{u_j\}$ is any orthonormal basis. Hence, the average singular value can be obtained directly from the joint even moments of the entries of $J_{\mathcal{N}}$. Both the quenched and annealed EVGP (see (7),(9))

entail that the average singular value for $J_\mathcal{N}$ equals 1, and we prove in Theorem 1 (specifically (11)) that even at finite depth and width the average singular value for $J_\mathcal{N}$ equals 1 for all the random ReLU nets we consider!

One can push this line of reasoning further. Namely, the singular values of any matrix $M$ are determined by the Stieltjes transform of the empirical distribution $\sigma_M$ of the eigenvalues of $M^T M$ :

$$S_M(z) = \int_\mathbb{R} \frac{d\sigma_M(x)}{z - x}, \qquad z \in \mathbb{C}\backslash\mathbb{R}.$$

Writing $(z - x)^{-1}$ as a power series in $z$ shows that $S_{J_\mathcal{N}}$ is determined by traces of powers of $J_\mathcal{N}^T J_\mathcal{N}$ and hence by the joint even moments of the entries of $J_\mathcal{N}$. We hope to estimate $S_{J_\mathcal{N}}(z)$ directly in future work.

### 3.2 Annealed Exploding and Vanishing Gradients

Fix a sequence of positive integers $n_0, n_1, \ldots$ For each $d \geq 1$ write $\mathcal{N}_d$ for the depth $d$ ReLU net with hidden layer widths $n_0, \ldots, n_d$ and random weights and biases (see Definition 1 below). As in (3), write $Z_{p,q}(d)$ for the partial derivative of the $q^{th}$ component of the output of $\mathcal{N}_d$ with respect to $p^{th}$ component of its input. We say that the family of architectures given by $\{n_0, n_1, \ldots\}$ *avoids the exploding and vanishing gradient problem in the annealed sense* if for each fixed input to $\mathcal{N}_d$ and every $p, q$ we have

$$\mathbb{E}\left[Z_{p,q}^2(d)\right] = 1, \quad \mathrm{Var}[Z_{p,q}^2(d)] = \Theta(1), \quad \sup_{d \geq 1} \mathbb{E}\left[Z_{p,q}^{2K}(d)\right] \ < \ \infty, \ \ \forall K \geq 3. \tag{7}$$

Here the expectation is over the weights and biases in $\mathcal{N}_d$. Architectures that avoid the EVGP in the annealed sense are ones where the typical magnitude of the partial derivatives $Z_{p,q}(d)$ have bounded (both above and below) fluctuations around a constant mean value. This allows for a reliable a priori selection of the learning rate $\lambda$ from (1) even for deep architectures. Our main result about the annealed EVGP is Theorem 1: a family of neural net architectures avoids the EVGP in the annealed sense if and only if

$$\sum_{j=1}^\infty \frac{1}{n_j} \ < \ \infty. \tag{8}$$

We prove in Theorem 1 that $\mathbb{E}\left[Z_{p,q}^{2K}(d)\right]$ is exponential in $\sum_{j \leq d} 1/n_j$ for every $K$.

### 3.3 Quenched Exploding and Vanishing Gradients

There is an important objection to defining the EVGP as in the previous section. Namely, if a neural net $\mathcal{N}$ suffers from the annealed EVGP, then it is impossible to choose an appropriate a priori learning rate $\lambda$ that works for a typical initialization. However, it may still be that for a typical realization of the weights and biases there is some choice of $\lambda$ (depending on the particular initialization), that works well for all (or most) trainable parameters in $\mathcal{N}$. To study whether this is the case, we must consider the variation of the $Z_{p,q}$'s across different $p, q$ in a fixed realization of weights and biases. This is the essence of the quenched EVGP.

To formulate the precise definition, we again fix a sequence of positive integers $n_0, n_1, \ldots$ and write $\mathcal{N}_d$ for a depth $d$ ReLU net with hidden layer widths $n_0, \ldots, n_d$. We write as in (4)

$$\widehat{\mathrm{Var}}\left[Z(d)^2\right] := \frac{1}{M}\sum_{m=1}^M Z_{p_m,q_m}(d)^4 - \left(\frac{1}{M}\sum_{m=1}^M Z_{p_m,q_m}(d)^2\right)^2, \qquad M = n_0 n_d$$

for the empirical variance of the squares all the entries $Z_{p,q}(d)$ of the input-output Jacobian of $\mathcal{N}_d$. We will say that the family of architectures given by $\{n_0, n_1, \ldots\}$ *avoids the exploding and vanishing gradient problem in the quenched sense* if

$$\mathbb{E}\left[Z_{p,q}(d)^2\right] = 1 \qquad \text{and} \qquad \mathbb{E}\left[\widehat{\mathrm{Var}}[Z(d)^2]\right] = \Theta(1). \tag{9}$$

Just as in the annealed case (7), the expectation $\mathbb{E}\left[\cdot\right]$ is with respect to the weights and biases of $\mathcal{N}$. In words, a neural net architecture suffers from the EVGP in the quenched sense if for a

typical realization of the weights and biases the empirical variance of the squared partial derivatives $\{Z^2_{p_m,q_m}\}$ is large.

Our main result about the quenched sense of the EVGP is Theorem 2. It turns out, at least for the ReLU nets we study, that a family of neural net architectures avoids the quenched EVGP if and only if it also avoids the annealed exploding and vanishing gradient problem (i.e. if (8) holds).

## 4   Acknowledgements

I thank Leonid Hanin for a number of useful conversations and for his comments on an early draft. I am also grateful to Jeffrey Pennington for pointing out an important typo in the proof of Theorem 3 and to David Rolnick for several helpful conversations and, specifically, for pointing out the relevance of the power-mean inequality for understanding $\beta$. Finally, I would like to thank several anonymous referees for their help in improving the exposition. One referee in particular raised concerns about the annealed and quenched definitions of the EVGP. Addressing these concerns resulted in the discussion in §3.1.

## 5   Notation and Main Results

### 5.1   Definition of Random Networks

To formally state our results, we first give the precise definition of the random networks we study. For every $d \geq 1$ and each $\mathbf{n} = (n_i)_{i=0}^d \in \mathbb{Z}_+^{d+1}$, write

$$\mathfrak{N}(\mathbf{n}, d) = \left\{ \begin{matrix} \text{fully connected feed-forward nets with ReLU activations,} \\ \text{depth } d, \text{ and whose } j^{th} \text{ hidden layer has width } n_j \end{matrix} \right\}.$$

The function $f_\mathcal{N}$ computed by $\mathcal{N} \in \mathfrak{N}(\mathbf{n}, d)$ is determined by a collection of weights and biases

$$\{w_{\alpha,\beta}^{(j)}, b_\beta^{(j)}, \quad 1 \leq \alpha \leq n_j, \; 1 \leq \beta \leq n_{j+1}, \; j = 0, \ldots, d-1\}.$$

Specifically, given an input

$$\text{Act}^{(0)} = \left( \text{Act}_i^{(0)} \right)_{i=1}^{n_0} \in \mathbb{R}^{n_0}$$

to $\mathcal{N}$, we define for every $j = 1, \ldots, d$

$$\text{act}_\beta^{(j)} = b_\beta^{(j)} + \sum_{\alpha=1}^{n_{j-1}} \text{Act}_\alpha^{(j-1)} w_{\alpha,\beta}^{(j)}, \qquad \text{Act}_\beta^{(j)} = \phi(\text{act}_\beta^{(j)}), \quad 1 \leq \beta \leq n_j. \tag{10}$$

The vectors $\text{act}^{(j)}$, $\text{Act}^{(j)}$ therefore represent the vectors of inputs and outputs of the neurons in the $j^{th}$ layer of $\mathcal{N}$. The function computed by $\mathcal{N}$ takes the form

$$f_\mathcal{N}\left( \text{Act}^{(0)} \right) = f_\mathcal{N}\left( \text{Act}^{(0)}, w_{\alpha,\beta}^{(j)}, b_\beta^{(j)} \right) = \text{Act}^{(d)}.$$

A random network is obtained by randomizing weights and biases.

**Definition 1** (Random Nets). *Fix $d \geq 1$, $\mathbf{n} = (n_0, \ldots, n_d) \in \mathbb{Z}_+^{d+1}$, and two collections of probability measures $\mu = \left( \mu^{(1)}, \ldots, \mu^{(d)} \right)$ and $\nu = \left( \nu^{(1)}, \ldots, \nu^{(d)} \right)$ on $\mathbb{R}$ such that*

*(i) $\mu^{(j)}, \nu^{(j)}$ are symmetric around $0$ for every $1 \leq j \leq d$.*

*(ii) the variance of $\mu^{(j)}$ is $2/(n_{j-1})$.*

*(iii) $\nu^{(j)}$ has no atoms.*

*A random net $\mathcal{N} \in \mathfrak{N}_{\mu,\nu}(\mathbf{n}, d)$ is obtained by requiring that the weights and biases for neurons at layer $j$ are drawn independently from $\mu^{(j)}, \nu^{(j)}$ :*

$$w_{\alpha,\beta}^{(j)} \sim \mu^{(j)}, \; b_\beta^{(j)} \sim \nu^{(j)} \qquad i.i.d.$$

**Remark 1.** *Condition (iii) is used when we apply Lemma 1 in the proof of Theorem 3. It can be removed under the restriction that $d \ll \exp\left( \sum_{j=1}^d n_j \right)$. Since this yields slightly messier but not meaningfully different results, we do not pursue this point.*

## 5.2 Results

Our main theoretical results are Theorems 1 and 3. They concern the statistics of the slopes of the functions computed by a random neural net in the sense of Definition 1. To state them compactly, we define for any probability measure $\mu$ on $\mathbb{R}$

$$\widetilde{\mu}_{2K} := \frac{\int_{\mathbb{R}} x^{2K} d\mu}{\left(\int_R x^2 d\mu\right)^K}, \qquad K \geq 0,$$

and, given a collection of probability measures $\{\mu^{(j)}\}_{j=1}^d$ on $\mathbb{R}$, set for any $K \geq 1$

$$\widetilde{\mu}_{2K,max} := \max_{1 \leq j \leq d} \widetilde{\mu}_{2K}^{(j)}.$$

We also continue to write $Z_{p,q}$ for the entries of the input-output Jacobian of a neural net (see (3)).

**Theorem 1.** *Fix $d \geq 1$ and a multi-index $\mathbf{n} = (n_0, \ldots, n_d) \in \mathbb{Z}_+^{d+1}$. Let $\mathcal{N} \in \mathfrak{N}_{\mu,\nu}(\mathbf{n}, d)$ be a random network as in Definition 1. For any fixed input to $\mathcal{N}$, we have*

$$\mathbb{E}\left[Z_{p,q}^2\right] = \frac{1}{n_0}. \tag{11}$$

*In contrast, the fourth moment of $Z_{p,q}(x)$ is exponential in $\sum_j \frac{1}{n_j}$ :*

$$\frac{2}{n_0^2} \exp\left(\frac{1}{2} \sum_{j=1}^{d-1} \frac{1}{n_j}\right) \leq \mathbb{E}\left[Z_{p,q}^4\right] \leq \frac{6\widetilde{\mu}_{4,max}}{n_0^2} \exp\left(6 \, \widetilde{\mu}_{4,max} \sum_{j=1}^{d-1} \frac{1}{n_j}\right). \tag{12}$$

*Moreover, there exists a constant $C_{K,\mu} > 0$ depending only on $K$ and the first $2K$ moments of the measures $\{\mu^{(j)}\}_{j=1}^d$ such that if $K < \min_{j=1}^{d-1}\{n_j\}$, then*

$$\mathbb{E}\left[Z_{p,q}^{2K}\right] \leq \frac{C_{K,\mu}}{n_0^K} \exp\left(C_{K,\mu} \sum_{j=1}^{d-1} \frac{1}{n_j}\right). \tag{13}$$

**Remark 2.** *In (11), (12), and (13), the bias distributions $\nu^{(j)}$ play no role. However, in the derivation of these relations, we use in Lemma 1 that $\nu^{(j)}$ has no atoms (see Remark 1). Also, the condition $K < \min_{j=1}^d\{n_{j-1}\}$ can be relaxed by allowing $K$ to violate this inequality a fixed finite number $\ell$ of times. This causes the constant $C_{K,\mu}$ to depend on $\ell$ as well.*

We prove Theorem 1 in Appendix B. The constant factor multiplying $\sum_j 1/n_j$ in the exponent on the right hand side of (12) is not optimal and can be reduced by a more careful analysis along the same lines as the proof of Theorem 1 given below. We do not pursue this here, however, since we are primarily interested in fixing $K$ and understanding the dependence of $\mathbb{E}\left[Z_{p,q}(x)^{2K}\right]$ on the widths $n_j$ and the depth $d$. Although we've stated Theorem 1 only for the even moments of $Z_{p,q}$, the same techniques will give analogous estimates for any mixed even moments $\mathbb{E}\left[Z_{p_1,q}^{2K_1} \cdots Z_{p_m,q}^{2K_m}\right]$ when $K$ is set to $\sum_m K_m$ (see Remark 3). In particular, we can estimate the mean of the empirical variance of gradients.

**Theorem 2.** *Fix $n_0, \ldots, n_d \in \mathbb{Z}_+$, and let $\mathcal{N}$ be a random fully connected depth $d$ ReLU net with hidden layer widths $n_0, \ldots, n_d$ and random weights and biases as in Definition 1. Write $M = n_0 n_d$ and write $\widehat{\mathrm{Var}}\left[Z^2\right]$ for the empirical variance of the squares $\{Z_{p_m,q_m}^2\}$ of all $M$ input-output neuron pairs as in (4). We have*

$$\mathbb{E}\left[\widehat{\mathrm{Var}}[Z^2]\right] \leq \left(1 - \frac{1}{M}\right) \frac{6\widetilde{\mu}_{4,max}}{n_0^2} \exp\left(6 \, \widetilde{\mu}_{4,max} \sum_{j=1}^{d-1} \frac{1}{n_j}\right) \tag{14}$$

*and*

$$\mathbb{E}\left[\widehat{\mathrm{Var}}[Z^2]\right] \geq \frac{1}{n_0^2}\left(1 - \frac{1}{M}\right)\left(1 - \eta + \frac{4\eta}{n_1}\left(\widetilde{\mu}_4^{(1)} - 1\right)e^{-\frac{1}{n_1}}\right) \exp\left(\frac{1}{2} \sum_{j=1}^{d-1} \frac{1}{n_j}\right) \tag{15}$$

*where*

$$\eta := \frac{\#\{m_1, m_2 \mid m_1 \neq m_2, \ q_{m_1} = q_{m_2}\}}{M(M-1)} = \frac{n_0 - 1}{n_0 n_d - 1}.$$

*Hence, the family $\mathcal{N}_d$ of ReLU nets avoids the exploding and vanishing gradient problem in the quenched sense if and only if*

$$\sum_{j=1}^{\infty} \frac{1}{n_j} < \infty.$$

We prove Theorem 2 in Appendix C. The results in Theorems 1 and 2 are based on *exact* expressions, given in Theorem 3, for the even moments $\mathbb{E}\left[Z_{p,q}(x)^{2K}\right]$ in terms only of the moments of the weight distributions $\mu^{(j)}$. To give the formal statement, we introduce the following notation. For any $\mathbf{n} = (n_i)_{i=0}^d$ and any $1 \leq p \leq n_0$, $1 \leq q \leq n_d$, we say that a path $\gamma$ from the $p^{th}$ input neuron to the $q^{th}$ output neuron in $\mathcal{N} \in \mathfrak{N}(\mathbf{n}, d)$ is a sequence

$$\{\gamma(j)\}_{j=0}^d, \qquad 1 \leq \gamma(j) \leq n_j, \quad \gamma(0) = p, \quad \gamma(d) = q,$$

so that $\gamma(j)$ represents a neuron in the $j^{th}$ layer of $\mathcal{N}$. Similarly, given any collection of $K \geq 1$ paths $\Gamma = (\gamma_k)_{k=1}^K$ that connect (possibly different) neurons in the input of $\mathcal{N}$ with neurons in its output and any $1 \leq j \leq d$, denote by

$$\Gamma(j) = \bigcup_{\gamma \in \Gamma} \{\gamma(j)\}$$

the neurons in the $j^{th}$ layer of $\mathcal{N}$ that belong to at least one element of $\Gamma$. Finally, for every $\alpha \in \Gamma(j-1)$ and $\beta \in \Gamma(j)$, denote by

$$|\Gamma_{\alpha,\beta}(j)| = \#\{\gamma \in \Gamma \mid \gamma(j-1) = \alpha, \ \gamma(j) = \beta\}$$

the number of paths in $\Gamma$ that pass through neuron $\alpha$ at layer $j-1$ and through neuron $\beta$ at layer $j$.

**Theorem 3.** *Fix $d \geq 1$ and $\mathbf{n} = (n_0, \ldots, n_d) \in \mathbb{Z}_+^{d+1}$. Let $\mathcal{N} \in \mathfrak{N}_{\mu,\nu}(\mathbf{n}, d)$ be a random network as in Definition 1. For every $K \geq 1$ and all $1 \leq p \leq n_0$, $1 \leq q \leq n_d$, we have*

$$\mathbb{E}\left[Z_{p,q}^{2K}\right] = \sum_{\Gamma} \prod_{j=1}^d C_j(\Gamma), \tag{16}$$

*where the sum is over ordered tuples $\Gamma = (\gamma_1, \ldots, \gamma_{2K})$ of paths in $\mathcal{N}$ from $p$ to $q$ and*

$$C_j(\Gamma) = \left(\frac{1}{2}\right)^{|\Gamma(j)|} \prod_{\substack{\alpha \in \Gamma(j-1) \\ \beta \in \Gamma(j)}} \mu_{|\Gamma_{\alpha,\beta}(j)|}^{(j)},$$

*where for every $r \geq 0$, the quantity $\mu_r^{(j)}$ denotes the $r^{th}$ moment of the measure $\mu^{(j)}$.*

**Remark 3.** *The expression (16) can be generalized to case of mixed even moments. Namely, given $m \geq 1$ and for each $1 \leq m \leq M$ integers $K_m \geq 0$ and $1 \leq p_m \leq n_0$ $1 \leq q_m \leq n_d$, we have*

$$\mathbb{E}\left[\prod_{m=1}^M Z_{p_m,q_m}(x)^{2K_m}\right] = \sum_{\Gamma} \prod_{j=1}^d C_j(\Gamma), \tag{17}$$

*where now the sum is over collections $\Gamma = (\gamma_1, \ldots, \gamma_{2K})$ of $2K = \sum_m 2K_m$ paths in $\mathcal{N}$ with exactly $2K_m$ paths from $p_m$ to $q_m$. The proof is identical up to the addition of several well-placed subscripts.*

See Appendix A for the proof of Theorem 3.

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
