[Supplementary Material]

## A  Proof of Theorem 3

We will use the following observation.

**Lemma 1.** *Suppose $X, w_1, \ldots, w_n$ are independent real-valued random variables whose distributions are symmetric around 0. Assume also that the distribution of $X$ has no atoms (i.e. $\mathbb{P}(X = x) = 0$ for all $x \in \mathbb{R}$), and fix any bounded positive function $\psi : \mathbb{R} \to \mathbb{R}_+$ with the property*

$$\psi(t) + \psi(-t) = 1. \tag{18}$$

*Then for any constants $a_1, \ldots, a_n \in \mathbb{R}$ and any non-negative integers $k_1, \ldots, k_n$ whose sum is even, we have*

$$\mathbb{E}\left[\prod_{j=1}^{n} w_j^{k_j} \psi(X + \sum_j w_j a_j)\right] = \frac{1}{2} \prod_{j=1}^{n} \mathbb{E}\left[w_j^{k_j}\right].$$

*Proof.* Using that $X \stackrel{d}{=} -X$, $w_j \stackrel{d}{=} -w_j$ and that $\sum_j k_j$ is even, we have

$$\mathbb{E}\left[\prod_{j=1}^{n} w_j^{k_j} \psi(X + \sum_j w_j a_j)\right] = \mathbb{E}\left[\prod_{j=1}^{n} w_j^{k_j} \psi(-(X + \sum_j w_j a_j))\right].$$

Averaging these two expressions we combine (18) with the fact that $X$ is independent of $\{w_j\}_{j=1}^{n}$ and its law has no atoms to obtain the desired result. s ☐

We now turn to the proof of Theorem 3. To this end, fix $d \geq 1$, a collection of positive integers $\mathbf{n} = (n_i)_{i=0}^{d}$, and let $\mathcal{N} \in \mathfrak{N}_{\mu,\nu}(\mathbf{n}, d)$. Let us briefly recall the notation for paths from §5.2. Given $1 \leq p \leq n_0$ and $1 \leq q \leq n_d$, we defined a path $\gamma$ from p to q to be a collection $\{\gamma(j)\}_{j=0}^{d}$ of neurons so that $\gamma(0) = p$, $\gamma(d) = q$, and $\gamma(j) \in \{1, \ldots, n_j\}$. The numbers $\gamma(j)$ should be thought of as neurons in the $j^{th}$ hidden layer of $\mathcal{N}$. Given such a collection, we obtain for each $j$ a weight

$$w_\gamma^{(j)} := w_{\gamma(j-1),\gamma(j)}^{(j)} \tag{19}$$

between each two consecutive neurons along the path $\gamma$. Our starting point is the expression

$$Z_{p,q} = \sum_{\substack{\text{paths } \gamma \\ \text{from p to q}}} \prod_{j=1}^{d} w_\gamma^{(j)} \mathbf{1}_{\{\text{act}_{\gamma(j)}^{(j)} > 0\}}, \tag{20}$$

where $\text{act}^{(j)}$ are defined as in (10). This expression is well-known and follows immediately form the chain rule (c.f. e.g. equation (1) in [CHM$^+$15]). We therefore have

$$Z_{p,q}^{2K} = \sum_{\substack{\text{paths } \gamma_1, \ldots, \gamma_{2K} \\ \text{from p to q}}} \prod_{j=1}^{d} \prod_{k=1}^{2K} w_{\gamma_k}^{(j)} \mathbf{1}_{\{\text{act}_{\gamma_k(j)}^{(j)} > 0\}}.$$

We will prove a slightly more general statement than in the formulation of Theorem 3. Namely, suppose $\Gamma = (\gamma_1, \ldots, \gamma_{2K})$ is any collection of paths from the input of $\mathcal{N}$ to the output (the paths are not required to have the same starting and ending neurons) such that for every $\beta \in \Gamma(d)$,

$$\#\{\gamma \in \Gamma \mid \gamma(d) = \beta\} \quad \text{is even.}$$

We will show that

$$\mathbb{E}\left[\prod_{j=1}^{d} \prod_{k=1}^{2K} w_{\gamma_k}^{(j)} \mathbf{1}_{\{\text{act}_{\gamma_k(j)}^{(j)} > 0\}}\right] = \prod_{j=1}^{d} \left(\frac{1}{2}\right)^{|\Gamma(j)|} \prod_{\substack{\alpha \in \Gamma(j-1) \\ \beta \in \Gamma(j)}} \mu_{|\Gamma_{\alpha,\beta}(j)|}^{(j)}. \tag{21}$$

To evaluate the expectation in (21), note that the computation done by $\mathcal{N}$ is a Markov chain with respect to the layers (i.e. given $\text{Act}^{(j-1)}$, the activations at layers $j, \ldots, d$ are independent of the weight and biases up to and including layer $j - 1$.) Hence, denoting by $\mathcal{F}_{\leq d-1}$ the sigma algebra

generated by the weight and biases up to and including layer $d-1$, the tower property for expectation and the Markov property yield

$$
\mathbb{E}\left[\prod_{j=1}^{d}\prod_{k=1}^{2K} w_{\gamma_k}^{(j)} \mathbf{1}_{\{\text{act}_{\gamma_k(j)}^{(j)}>0\}}\right]
$$

$$
= \mathbb{E}\left[\prod_{j=1}^{d-1}\prod_{k=1}^{2K} w_{\gamma_k}^{(j)} \mathbf{1}_{\{\text{act}_{\gamma_k(j)}^{(j)}>0\}} \mathbb{E}\left[\prod_{k=1}^{2K} w_{\gamma_k}^{(d)} \mathbf{1}_{\{\text{act}_{\gamma_k(d)}^{(d)}>0\}} \mid \mathcal{F}_{\leq d-1}\right]\right]
$$

$$
= \mathbb{E}\left[\prod_{j=1}^{d-1}\prod_{k=1}^{2K} w_{\gamma_k}^{(j)} \mathbf{1}_{\{\text{act}_{\gamma_k(j)}^{(j)}>0\}} \mathbb{E}\left[\prod_{k=1}^{2K} w_{\gamma_k}^{(d)} \mathbf{1}_{\{\text{act}_{\gamma_k(d)}^{(d)}>0\}} \mid \text{Act}^{d-1}\right]\right]. \tag{22}
$$

Next, observe that for each $1 \leq j \leq d$, conditioned on $\text{Act}^{(j-1)}$, the families of random variables $\{w_{\alpha,\beta}^{(j)}, \text{act}_{\beta}^{(j)}\}_{\alpha=1}^{n_{j-1}}$ are independent for different $\beta$. For $j=d$ this implies

$$
\mathbb{E}\left[\prod_{k=1}^{2K} w_{\gamma_k}^{(d)} \mathbf{1}_{\{\text{act}_{\gamma_k(d)}^{(d)}>0\}} \mid \text{Act}^{(d-1)}\right] = \prod_{\beta \in \Gamma(d)} \mathbb{E}\left[\prod_{\substack{k=1 \\ \gamma_k(d)=\beta}}^{2K} w_{\gamma_k}^{(d)} \mathbf{1}_{\{\text{act}_{\gamma_k(d)}^{(d)}>0\}} \mid \text{Act}^{(d-1)}\right]. \tag{23}
$$

Consider the decomposition

$$
\text{act}_{\beta}^{(d)} = \text{act}_{\Gamma,\beta}^{(d)} + \widehat{\text{act}}_{\Gamma,\beta}^{(d)}, \tag{24}
$$

where

$$
\text{act}_{\Gamma,\beta}^{(d)} := \sum_{\alpha \in \Gamma(d-1)} \text{Act}_{\alpha}^{(d-1)} w_{\alpha,\beta}^{(d)}
$$

$$
\widehat{\text{act}}_{\Gamma,\beta}^{(d)} = \text{act}_{\beta}^{(d)} - \text{act}_{\Gamma,\beta}^{(d)} = b_{\beta}^{(d)} + \sum_{\alpha \notin \Gamma(d-1)} \text{Act}_{\alpha}^{(d-1)} w_{\alpha,\beta}^{(d)}.
$$

Let us make several observations about $\widehat{\text{act}}_{\Gamma,\beta}^{(d)}$ and $\text{act}_{\Gamma,\beta}^{(d)}$ *when conditioned on* $\text{Act}^{(d-1)}$. First, the conditioned random variable $\widehat{\text{act}}_{\Gamma,\beta}^{(d-1)}$ is independent of the conditioned random variable $\text{act}_{\Gamma,\beta}^{(d-1)}$. Second, the distribution of $\widehat{\text{act}}_{\Gamma,\beta}^{(d)}$ conditioned on $\text{Act}^{(d-1)}$ is symmetric around 0. Third, since we assumed that the bias distributions $\nu^{(j)}$ for $\mathcal{N}$ have no atoms, the conditional distribution of $\widehat{\text{act}}_{\Gamma,\beta}^{(d)}$ also has no atoms. Fourth, $\text{act}_{\Gamma,\beta}^{(d-1)}$ is a linear combination of the weights $\{w_{\alpha,\beta}^{(d)}\}_{\alpha \in \Gamma(j-1)}$ with given coefficients $\{\text{Act}_{\alpha}^{(d-1)}\}_{\alpha \in \Gamma(j-1)}$. Since the weight distributions $\mu^{(j)}$ for $\mathcal{N}$ are symmetric around 0, the above five observations, together with (24) allow us to apply Lemma 1 and to conclude that

$$
\mathbb{E}\left[\prod_{k=1}^{2K} w_{\gamma_k}^{(j)} \mathbf{1}_{\{\text{act}_{\gamma_k(d)}^{(d)}>0\}} \mid \text{Act}^{(d-1)}\right] = \left(\frac{1}{2}\right)^{|\Gamma(d)|} \prod_{\substack{\beta \in \Gamma(d) \\ \alpha \in \Gamma(d-1)}} \mu_{|\Gamma_{\alpha,\beta}(d)|}^{(d)}. \tag{25}
$$

Combining this with (22) yields

$$
\mathbb{E}\left[\prod_{j=1}^{d}\prod_{k=1}^{2K} w_{\gamma_k}^{(j)} \mathbf{1}_{\{\text{act}_{\gamma_k(j)}^{(j)}>0\}}\right]
$$

$$
= \mathbb{E}\left[\prod_{j=1}^{d-1}\prod_{k=1}^{2K} w_{\gamma_k}^{(j)} \mathbf{1}_{\{\text{act}_{\gamma_k(j)}^{(j)}>0\}}\right] \left(\frac{1}{2}\right)^{|\Gamma(d)|} \prod_{\substack{\beta \in \Gamma(d) \\ \alpha \in \Gamma(d-1)}} \mu_{|\Gamma_{\alpha,\beta}(d)|}^{(d)}.
$$

To complete the argument, we must consider two cases. First, recall that by assumption, for every $\beta \in \Gamma(d)$, the number of $\gamma \in \Gamma$ for which $\gamma(d) = \beta$ is even. If for every $j \leq d$ and each $\alpha \in \Gamma(j-1)$

the number of $\gamma \in \Gamma$ passing through $\alpha$ is even, then we may repeat the preceding argument to directly obtain (21). Otherwise, we apply this argument until we reach $\alpha \in \Gamma(j-1)$, $\beta \in \Gamma(j)$ so that the number $|\Gamma_{\alpha,\beta}(j)|$ of paths in $\Gamma$ that pass through $\alpha$ and $\beta$ is odd. In this case, the right hand side of (22) vanishes since the measure $\mu^{(d)}$ is symmetric around 0 and thus has vanishing odd moments. Relation (21) therefore again holds since in this case both sides are 0. This completes the proof of Theorem 3. $\qquad\square$

## B   Proof of Theorem 1

In this section, we use Theorem 3 to prove Theorem 1. Let us first check (11). According to Theorem 3, we have

$$\mathbb{E}\left[Z_{p,q}^2\right] = \sum_{\substack{\Gamma=(\gamma_1,\gamma_2) \\ \text{paths from p to q}}} \prod_{j=1}^{d} \left(\frac{1}{2}\right)^{|\Gamma(j)|} \prod_{\substack{\alpha\in\Gamma(j-1) \\ \beta\in\Gamma(j)}} \mu^{(j)}_{|\Gamma_{\alpha,\beta}(d)|}.$$

Note that since $\mu$ is symmetric around 0, we have that $\mu_1 = 0$. Thus, the terms where $\gamma_1 \neq \gamma_2$ vanish. Using $\mu_2^{(j)} = \frac{2}{n_{j-1}}$, we find

$$\mathbb{E}\left[Z_{p,q}^2\right] = \sum_{\substack{\text{paths } \gamma \\ \text{from p to q}}} \prod_{j=1}^{d} \frac{1}{2}\cdot\frac{2}{n_{j-1}} = \frac{1}{n_0},$$

as claimed. We now turn to proving (12). Using Theorem 3, we have

$$\mathbb{E}\left[Z_{p,q}^4\right] = \sum_{\substack{\Gamma=(\gamma_k)_{k=1}^4 \\ \text{paths from p to q}}} \prod_{j=1}^{d} \left(\frac{1}{2}\right)^{|\Gamma(j)|} \prod_{\substack{\beta\in\Gamma(j) \\ \alpha\in\Gamma(j-1)}} \mu^{(j)}_{|\Gamma_{\alpha,\beta}(j)|}$$

$$= \sum_{\substack{\Gamma=(\gamma_k)_{k=1}^4 \\ \text{paths from p to q} \\ |\Gamma_{\alpha,\beta}(j)| \text{ even } \forall \alpha,\beta}} \prod_{j=1}^{d} \left( \frac{\mu_4^{(j)}}{2}\mathbf{1}_{\left\{\substack{|\Gamma(j-1)|=1 \\ |\Gamma(j)|=1}\right\}} + \frac{\left(\mu_2^{(j)}\right)^2}{2}\mathbf{1}_{\left\{\substack{|\Gamma(j-1)|=2 \\ |\Gamma(j)|=1}\right\}} + \frac{\left(\mu_2^{(j)}\right)^2}{4}\mathbf{1}_{\{|\Gamma(j)|=2\}} \right),$$

where we have used that $\mu_1^{(j)} = \mu_3^{(j)} = 0$. Fix $\bar{\Gamma} = (\gamma_k)_{k=1}^4$. Note that $\bar{\Gamma}$ gives a non-zero contribution to $\mathbb{E}\left[Z_{p,q}^4\right]$ only if

$$\left|\bar{\Gamma}_{\alpha,\beta}(j)\right| \text{ is even,} \qquad \forall j, \alpha, \beta.$$

For each such $\bar{\Gamma}$, we have $\left|\Gamma(j)\right| \in \{1, 2\}$ for every $j$. Hence, for every $\bar{\Gamma}$ that contributes a non-zero term in the expression above for $\mathbb{E}\left[Z_{p,q}^4\right]$, we may find a collection of two paths $\Gamma = (\gamma_1, \gamma_2)$ from p to q such that

$$\bar{\Gamma}(j) = \Gamma(j), \qquad \left|\bar{\Gamma}_{\alpha,\beta}(j)\right| = 2\left|\Gamma_{\alpha,\beta}(j)\right|, \quad \forall j, \alpha, \beta.$$

We can thus write $\mathbb{E}\left[Z_{p,q}^4\right]$ as

$$\sum_{\substack{\Gamma=(\gamma_1,\gamma_2) \\ \text{paths from p to q}}} A(\Gamma) \prod_{j=1}^{d} \left( \frac{\mu_4^{(j)}}{2}\mathbf{1}_{\left\{\substack{|\Gamma(j-1)|=1 \\ |\Gamma(j)|=1}\right\}} + \frac{\left(\mu_2^{(j)}\right)^2}{2}\mathbf{1}_{\left\{\substack{|\Gamma(j-1)|=2 \\ |\Gamma(j)|=1}\right\}} + \frac{\left(\mu_2^{(j)}\right)^2}{4}\mathbf{1}_{\{|\Gamma(j)|=2\}} \right), \tag{26}$$

where we introduced

$$A(\Gamma) := \frac{\#\left\{\bar{\Gamma} = (\bar{\gamma}_k)_{k=1}^4, \; \bar{\gamma}_k \text{ path from p to q} \; \Big|\; \substack{\forall j,\alpha,\beta, \; \Gamma(j)=\bar{\Gamma}(j) \\ 2|\Gamma_{\alpha,\beta}(j)|=|\bar{\Gamma}_{\alpha,\beta}(j)|}\right\}}{\#\left\{\bar{\Gamma} = (\bar{\gamma}_k)_{k=1}^2, \; \bar{\gamma}_k \text{ path from p to q} \; \Big|\; \substack{\forall j,\alpha,\beta, \; \Gamma(j)=\bar{\Gamma}(j) \\ |\Gamma_{\alpha,\beta}(j)|=|\bar{\Gamma}_{\alpha,\beta}(j)|}\right\}}, \tag{27}$$

which we now evaluate.

**Lemma 1.** *For each $\Gamma = (\gamma_k)_{k=1}^2$ with $\gamma_k$ paths from p to q, we have*

$$A(\Gamma) = 3^{\#\{j\,|\,|\Gamma(j-1)|=1,\,|\Gamma(j)|=2\}} = 3^{\#\{j\,|\,|\Gamma(j-1)|=2,\,|\Gamma(j)|=1\}}. \qquad (28)$$

*Proof.* We begin by checking the first equality in (28) by induction on $d$. Fix $\Gamma = (\gamma_1, \gamma_2)$. When $d = 1$, we have $|\Gamma(0)| = |\Gamma(1)| = 1$. Hence $\gamma_1 = \gamma_2$ and $A(\Gamma) = 1$ since both the numerator and denominator on the right hand side of (27) equal 1. The right hand side of (28) is also 1 since $|\Gamma(j)| = 1$ for every $j$. This completes the base case. Suppose now that $D \geq 2$, and we have proved (28) for all $d \leq D - 1$. Let

$$j_* := \min\{j = 1, \ldots, d \,|\, |\Gamma(j)| = 1\}.$$

If $j_* = 1$, then we are done by the inductive hypothesis. Otherwise, there are two choices of $\bar{\Gamma} = \{\bar{\gamma}_k\}_{k=1}^2$ for which

$$\Gamma(j) = \bar{\Gamma}(j), \qquad |\Gamma_{\alpha,\beta}(j)| = |\bar{\Gamma}_{\alpha,\beta}(j)|, \qquad j \leq j_*.$$

These choices correspond to the two permutations of $\{\gamma_k\}_{k=1}^2$. Similarly, there are 6 choices of $\bar{\Gamma} = \{\bar{\gamma}_k\}_{k=1}^4$ for which

$$\Gamma(j) = \bar{\Gamma}(j), \qquad 2|\Gamma_{\alpha,\beta}(j)| = |\bar{\Gamma}_{\alpha,\beta}(j)|, \qquad j \leq j_*.$$

The six choices correspond to selecting one of two choices for $\gamma_1(1)$ and three choices of an index $k = 2, 3, 4$ so that $\gamma_k(j)$ coincides with $\gamma_1(j)$ for each $j \leq j_*$. If $j_* = d$, we are done. Otherwise, we apply the inductive hypothesis to paths from $\Gamma(j_*)$ to $\Gamma(d)$ to complete the proof of the first equality in (28). The second equality in (28) follows from the observation that since $|\Gamma(0)| = |\Gamma(d)| = 1$, the number of $j \in \{1, \ldots, d\}$ for which $|\Gamma(j-1)| = 1$, $|\Gamma(j)| = 2$ must equal the number of $j$ for which $|\Gamma(j-1)| = 2$, $|\Gamma(j)| = 1$. $\qquad \square$

Combining (26) with (28), we may write $\mathbb{E}\left[Z_{p,q}^4\right]$ as

$$\sum_{\substack{\Gamma=(\gamma_1,\gamma_2) \\ \text{paths from p to q}}} \prod_{j=1}^d \left[\frac{\mu_4^{(j)}}{2}\mathbf{1}_{\left\{\substack{|\Gamma(j-1)|=1 \\ |\Gamma(j)|=1}\right\}} + \frac{3}{2}\left(\mu_2^{(j)}\right)^2\mathbf{1}_{\left\{\substack{|\Gamma(j-1)|=2 \\ |\Gamma(j)|=1}\right\}} + \frac{\left(\mu_2^{(j)}\right)^2}{4}\mathbf{1}_{\{|\Gamma(j)|=2\}}\right]. \quad (29)$$

Observe that since $\mu_2^{(j)} = 2/n_{j-1}$, we have

$$\left(\#\{\Gamma = (\gamma_k)_{k=1}^2 \text{ paths from p to q}\}\right)^{-1} = \prod_{j=1}^{d-1}\frac{1}{n_j^2} = n_0^2 \cdot \prod_{j=1}^d \left(\mu_2^{(j)}\right)^2 / 4.$$

Hence,

$$\mathbb{E}\left[Z_{p,q}^4\right] = \frac{1}{n_0^2}\mathbb{E}\left[X_d\left(\gamma_1, \gamma_2\right)\right],$$

where the expectation on the right hand side is over the uniform measure on paths $(\gamma_1, \gamma_2)$ from the input of $\mathcal{N}$ to the output conditioned on $\gamma_1(0) = \gamma_2(0) = p$ and $\gamma_1(d) = \gamma_2(d) = q$, and

$$X_d(\gamma_1, \gamma_2) := \prod_{j=1}^d \left(2\widetilde{\mu}_4 \cdot \mathbf{1}_{\left\{\substack{|\Gamma(j-1)|=1 \\ |\Gamma(j)|=1}\right\}} + 6 \cdot \mathbf{1}_{\left\{\substack{|\Gamma(j-1)|=2 \\ |\Gamma(j)|=1}\right\}} + \mathbf{1}_{\{|\Gamma(j)|=2\}}\right), \quad \Gamma = (\gamma_1, \gamma_2).$$

We now obtain the upper and lower bounds in (12) on $\mathbb{E}\left[Z_{p,q}^4\right]$ in similar ways. In both cases, we use the observation that the number of $\Gamma = (\gamma_1, \gamma_2)$ for which $|\Gamma(j)| = 1$ for exactly $k$ values of $1 \leq j \leq d-1$ is

$$\prod_{j=1}^{d-1} n_j \cdot \sum_{\substack{I \subseteq \{1,\ldots,d-1\} \\ |I|=d-1-k}} \prod_{j \in I}(n_j - 1).$$

The value of $X_d$ corresponding to every such path is at least $2^{k+1}$ since $\widetilde{\mu}_4^{(j)} \geq 1$ for every $j$ and is at most $6\widetilde{\mu}_{4,max}$ for the same reason. Therefore, using that for all $\varepsilon \in [0, 1]$, we have

$$\log(1 + \varepsilon) \geq \frac{\varepsilon}{2},$$

we obtain

$$\mathbb{E}\left[X_d\right] \geq \frac{1}{\prod_{j=1}^{d-1} n_j} \sum_{k=0}^{d-1} 2^{k+1} \sum_{\substack{I \subseteq \{1,\ldots,d-1\} \\ |I|=d-1-k}} \prod_{j \in I}(n_j - 1)$$

$$= 2 \sum_{k=0}^{d-1} \sum_{\substack{I \subseteq \{1,\ldots d-1\} \\ |I|=d-1-k}} \prod_{j \notin I}\left(\frac{2}{n_j}\right) \prod_{j \in I}\left(1 + \frac{1}{n_j}\right)$$

$$= 2 \prod_{j=1}^{d-1}\left(1 + \frac{1}{n_j}\right) \geq 2 \exp\left(\frac{1}{2} \sum_{j=1}^{d-1} \frac{1}{n_j}\right).$$

This completes the proof of the lower bound. The upper bound is obtained in the same way:

$$\mathbb{E}\left[X_d\right] \leq \frac{1}{\prod_{j=1}^{d-1} n_j} \sum_{k=0}^{d-1} (6\widetilde{\mu}_{4,max})^{k+1} \sum_{\substack{I \subseteq \{1,\ldots,d-1\} \\ |I|=d-1-k}} \prod_{j \in I}(n_j - 1)$$

$$= 6\widetilde{\mu}_{4,max} \prod_{j=1}^{d-1}\left(1 + \frac{6\widetilde{\mu}_{4,max}}{n_j}\right) \leq 6\widetilde{\mu}_{4,max} \exp\left(6\widetilde{\mu}_{4,max} \sum_{j=1}^{d-1} \frac{1}{n_j}\right). \qquad (30)$$

The upper bounds for $\mathbb{E}\left[Z_{p,q}^{2K}\right]$ for $K \geq 3$ are obtained in essentially the same way. Namely, we return to the expression for $\mathbb{E}\left[Z_{p,q}^{2K}\right]$ provided by Theorem 3:

$$\mathbb{E}\left[Z_{p,q}^{2K}\right] = \sum_{\substack{\Gamma = \{\gamma_k\}_{k=1}^{2K} \\ \gamma_k \text{ paths from p to q}}} \prod_{j=1}^{d}\left(\frac{1}{2}\right)^{|\Gamma(j)|} \prod_{\substack{\beta \in \Gamma(j) \\ \alpha \in \Gamma(j-1)}} \mu_{|\Gamma_{\alpha,\beta}(j)|}^{(j)}.$$

As with the second and fourth moment computations, we note that $\mu_{|\Gamma_{\alpha,\beta}(j)|}$ vanishes unless each $|\Gamma_{\alpha,\beta}(j)|$ is even. Hence, as with (29), we may write

$$\mathbb{E}\left[Z_{p,q}^{2K}\right] = \sum_{\substack{\Gamma = \{\gamma_k\}_{k=1}^{K} \\ \gamma_k \text{ paths from p to q}}} A_K(\Gamma) \prod_{j=1}^{d}\left(\frac{1}{2}\right)^{|\Gamma(j)|} \prod_{\substack{\beta \in \Gamma(j) \\ \alpha \in \Gamma(j-1)}} \mu_{2|\Gamma_{\alpha,\beta}(j)|}^{(j)}, \qquad (31)$$

where $A_K(\Gamma)$ is the analog of $A(\Gamma)$ from (27). The same argument as in Lemma 1 shows that

$$A_K(\Gamma) \leq \left(\frac{(2K)!}{K!}\right)^{\#\{1 \leq j \leq d \,|\, |\Gamma(j)| < K\}}.$$

Combining this with

$$\left(\#\{\Gamma = (\gamma_k)_{k=1}^{K} \text{ paths from p to q}\}\right)^{-1} = \prod_{j=1}^{d-1} \frac{1}{n_j^K} = n_0^K \cdot \prod_{j=1}^{d}\left(\mu_2^{(K)}\right)^2 / 2^K,$$

which is precisely the weight in (31) assigned to collections $\Gamma$ with $|\Gamma(j)| = K$ for every $1 \leq j \leq d-1$, yields

$$\mathbb{E}\left[Z_{p,q}^{2K}\right] \leq \frac{1}{n_0^K} \mathbb{E}\left[X_d\left(\gamma_1,\ldots,\gamma_K\right) \mid \gamma_k(0) = p, \ \gamma_k(d) = q\right],$$

where the expectation is over uniformly chosen collections $\Gamma = (\gamma_1,\ldots,\gamma_K)$ of paths from the input to the output of $\mathcal{N}$ and

$$X_d(\Gamma) = \prod_{j=1}^{d} 2^{K-|\Gamma(j)|} \frac{(2K)!}{K!} \prod_{\substack{\alpha \in \Gamma(j-1) \\ \beta \in \Gamma(j) \\ |\Gamma(j)| < K}} \mu_{2|\Gamma_{\alpha,\beta}(j)|}^{(j)}.$$

To complete the proof of the upper bound for $\mathbb{E}\left[Z_{p,q}^{2K}\right]$ we now proceed just as the upper bound for the $4^{th}$ moment computation. That is, given $K < \min\{n_j\}$, the number of collections of paths $\Gamma = (\gamma_k)_{k=1}^K$ which $|\Gamma(j)| < K$ for exactly $m$ values of $j$ is bounded above by

$$\prod_{j=1}^{d-1} n_j^{K-1} \sum_{\substack{I \subseteq \{1,\ldots,d-1\} \\ |I|=d-1-m}} \prod_{j \in I} (n_j - K).$$

The value of $X_d$ on each such collection is at most $(C_K)^m$, where $C_K = 2^{K-1} \frac{(2K)!}{K!}$ is a large but fixed constant. Hence, just as in (30),

$$\mathbb{E}\left[X_d(\Gamma)\right] \le C_K \exp\left(C_K \sum_{j=1}^{d-1} \frac{1}{n_j}\right).$$

This completes the proof of Theorem 1. $\qquad\qquad\qquad\qquad\qquad\qquad\qquad\qquad\square$

## C  Proof of Theorem 2

We have

$$\widehat{\mathrm{Var}[Z^2]} = \frac{1}{M}\left(1 - \frac{1}{M}\right) \sum_{m=1}^M Z_{p_m,q_m}^4 - \frac{1}{M^2} \sum_{m_1 \neq m_2} Z_{p_{m_1},q_{m_1}}^2 Z_{p_{m_2},q_{m_2}}^2. \tag{32}$$

Fixing $p, q$ and using that the second sum in the previous line has $M(M-1)$ terms, we have

$$-\frac{1}{M^2} \sum_{m_1 \neq m_2} Z_{p_{m_1},q_{m_1}}^2 Z_{p_{m_2},q_{m_2}}^2 = \frac{1}{M^2} \sum_{m_1 \neq m_2} \left(Z_{p,q}^4 - Z_{p_{m_1},q_{m_1}}^2 Z_{p_{m_2},q_{m_2}}^2\right) + \left(1 - \frac{1}{M}\right) Z_{p,q}^4.$$

Hence, using that $\mathbb{E}\left[Z_{p,q}^4\right]$ is independent of the particular values of $p, q$, we fix some $p, q$ and write

$$\mathbb{E}\left[\widehat{\mathrm{Var}[Z^2]}\right] = \frac{1}{M^2} \sum_{m_1 \neq m_2} \mathbb{E}\left[Z_{p,q}^4\right] - \mathbb{E}\left[Z_{p_{m_1},q_{m_1}}^2 Z_{p_{m_2},q_{m_2}}^2\right]. \tag{33}$$

To estimate the difference in this sum, we use Theorem 3 to obtain

$$\mathbb{E}\left[Z_{p,q}^4\right] = \sum_{\substack{\Gamma=(\gamma_k)_{k=1}^4 \\ \gamma_k: p \to q}} \prod_{j=1}^d \left(\frac{1}{2}\right)^{|\Gamma(j)|} \prod_{\alpha,\beta} \mu_{|\Gamma_{\alpha,\beta}(j)|}^{(j)} = \sum_{\substack{\Gamma=(\gamma_k)_{k=1}^4 \\ \gamma_k: p \to q}} \prod_{j=1}^d C_j(\Gamma) \tag{34}$$

$$\mathbb{E}\left[Z_{p_1,q_1}^2 Z_{p_2,q_2}^2\right] = \sum_{\substack{\bar{\Gamma}=(\gamma_k)_{k=1}^4 \\ \gamma_1,\gamma_2: p_1 \to q_1 \\ \gamma_3,\gamma_4: p_2 \to q_2}} \prod_{j=1}^d \left(\frac{1}{2}\right)^{|\bar{\Gamma}(j)|} \prod_{\alpha,\beta} \mu_{|\bar{\Gamma}_{\alpha,\beta}(j)|}^{(j)} = \sum_{\substack{\bar{\Gamma}=(\gamma_k)_{k=1}^4 \\ \gamma_1,\gamma_2: p_1 \to q_1 \\ \gamma_3,\gamma_4: p_2 \to q_2}} \prod_{j=1}^d C_j(\bar{\Gamma}). \tag{35}$$

Note that since the measures $\mu^{(j)}$ of the weights are symmetric around zero, their odd moments vanish and hence the only non-zero terms in (34) and (35) are those for which

$$|\Gamma(j)|, \left|\bar{\Gamma}(j)\right| \in \{1, 2\}, \quad |\Gamma_{\alpha,\beta}(j)|, \left|\bar{\Gamma}_{\alpha,\beta}(j)\right| \in \{2, 4\}, \qquad \forall j, \alpha, \beta.$$

Further, observe that each path $\gamma$ from some fixed input neuron to some fixed output vertex is determined uniquely by the sequence of hidden neurons $\gamma(j) \in \{1, \ldots, n_j\}$ through which it passes for $j = 1, \ldots, d-1$. Therefore, we may identify each collection of paths $\Gamma = (\gamma_k)_{k=1}^4$ in the sum (34) with a unique collection of paths $\bar{\Gamma} = (\bar{\gamma}_k)_{k=1}^4$ in (35) by asking that $\gamma_k(j) = \bar{\gamma}_k(j)$ for each $k$ and all $1 \le j \le d-1$. Observe further that under this bijection,

$$j \neq 1, d \quad \Rightarrow \quad C_j(\Gamma) = C_j(\bar{\Gamma}). \tag{36}$$

For $j = 1, d$, the terms $C_j(\Gamma)$ and $C_j(\bar{\Gamma})$ are related as follows:

$$C_1(\Gamma) = C_1(\bar{\Gamma}) \left( \mathbf{1}_{\{|\Gamma(1)|=2\}} + \widetilde{\mu}_4^{(1)} \cdot \mathbf{1}_{\{|\Gamma(1)|=1\}} \right) \tag{37}$$

$$C_d(\Gamma) = C_d(\bar{\Gamma}) \left( \mathbf{1}_{\{|\bar{\Gamma}(d)|=1\}} + 2 \cdot \mathbf{1}_{\left\{ \substack{|\bar{\Gamma}(d)|=2 \\ |\Gamma(d-1)|=2} \right\}} + 2\widetilde{\mu}_4^{(d)} \cdot \mathbf{1}_{\left\{ \substack{|\bar{\Gamma}(d)|=2 \\ |\Gamma(d-1)|=1} \right\}} \right). \tag{38}$$

We consider two cases: (i) $q_{m_1} \neq q_{m_2}$ (i.e. $\left| \bar{\Gamma}(d) \right| = 2$) and (ii) $q_{m_1} = q_{m_2}$ (i.e. $\left| \bar{\Gamma}(d) \right| = 1$ and $p_{m_1} \neq p_{m_2}$). In case (i), we have

$$C_1(\Gamma) = C_1(\bar{\Gamma}) \left( \mathbf{1}_{\{|\Gamma(1)|=2\}} + \widetilde{\mu}_4^{(1)} \mathbf{1}_{\{|\Gamma(1)|=1\}} \right) \geq C_1(\bar{\Gamma}) \qquad \text{and} \qquad C_d(\Gamma) \geq 2C_d(\bar{\Gamma}).$$

Hence, using (37) and (38), we find that in case (i)

$$q_{m_1} \neq q_{m_2} \qquad \Rightarrow \qquad \mathbb{E}\left[ Z_{p_m,q_m}^4 \right] \geq 2\mathbb{E}\left[ Z_{p_{m_1},q_{m_1}}^2 Z_{p_{m_2},q_{m_2}}^2 \right].$$

In case (i) we therefore find

$$\mathbb{E}\left[ Z_{p_m,q_m}^4 \right] - \mathbb{E}\left[ Z_{p_{m_1},q_{m_1}}^2 \right] \geq \mathbb{E}\left[ Z_{p_{m_1},q_{m_1}}^2 \right] \geq \frac{1}{n_0^2} \exp\left( \frac{1}{2} \sum_{j=1}^{d-1} \frac{1}{n_j} \right), \tag{39}$$

where the last estimate is proved by the same argument as the relation (12) in Theorem 1. To obtain the analogous lower bound for case (ii), we write $q = q_{m_1} = q_{m_2}$, $p_{m_1} \neq p_{m_2}$. In this case, combining (36) with (38), we have

$$C_j(\Gamma) = C_j(\bar{\Gamma}) \quad j = 2, \ldots, d.$$

Moreover, continuing to use the bijection between $\Gamma$ and $\bar{\Gamma}$ above, (37) yields in this case

$$C_1(\bar{\Gamma}) = \begin{cases} \frac{1}{\widetilde{\mu}_4^{(1)}} C_1(\Gamma) & , \qquad \text{if } |\Gamma(1)| = 1 \\ 0 & , \qquad \text{if } |\Gamma(1)| = 2 \end{cases}.$$

Hence, $\mathbb{E}\left[ Z_{p,q}^4 \right] - \mathbb{E}\left[ Z_{p_1,q}^2 Z_{p_2,q}^2 \right]$ becomes

$$\sum_{\substack{\Gamma=(\gamma_k)_{k=1}^4 \\ \gamma_k : p \to q}} (C_1(\Gamma) - C_1(\bar{\Gamma})) \prod_{j=2}^{d} C_j(\Gamma) \;=\; \left( 1 - \frac{1}{\widetilde{\mu}_4^{(1)}} \right) \sum_{\substack{\Gamma=(\gamma_k)_{k=1}^4 \\ \gamma_k : p \to q \\ |\Gamma(1)|=1}} \prod_{j=1}^{d} C_j(\Gamma).$$

Using that if $|\Gamma(0)| = |\Gamma(1)| = 1$, then

$$C_1(\Gamma) = \frac{\mu_4^{(1)}}{2} = \frac{2\widetilde{\mu}_4^{(1)}}{n_0^2},$$

we find

$$\mathbb{E}\left[ Z_{p,q}^4 \right] - \mathbb{E}\left[ Z_{p_1,q}^2 Z_{p_2,q}^2 \right] = \frac{2}{n_0^2} \left( \widetilde{\mu}_4^{(1)} - 1 \right) \sum_{\substack{\Gamma=(\gamma_k)_{k=1}^4 \\ \gamma_k : p \to q \\ |\Gamma(1)|=1}} \prod_{j=2}^{d} C_j(\Gamma). \tag{40}$$

Writing $\widehat{p}$ for any neuron in the first hidden layer of $\mathcal{N}$, we rewrite the sum in the previous line as

$$\sum_{\substack{\Gamma=(\gamma_k)_{k=1}^4 \\ \gamma_k : p \to q \\ |\Gamma(1)|=1}} \prod_{j=2}^{d} C_j(\Gamma) = n_1 \sum_{\substack{\Gamma=(\gamma_k)_{k=1}^4 \\ \gamma_k : \widehat{p} \to q}} \prod_{j=2}^{d} C_j(\Gamma) = n_1 \mathbb{E}\left[ Z_{\widehat{p},q}^4 \right],$$

where the point is now that we are considering paths only from $\widehat{p}$ to $q$. According to (12) from Theorem 1, we have

$$\mathbb{E}\left[ Z_{\widehat{p},q}^4 \right] \geq \frac{2}{n_1^2} \exp\left( \frac{1}{2} \sum_{j=2}^{d-1} \frac{1}{n_j} \right).$$

Combining this with (40) yields

$$\mathbb{E}\left[Z_{p,q}^4\right] - \mathbb{E}\left[Z_{p_1,q}^2 Z_{p_2,q}^2\right] \geq \frac{4}{n_0^2 n_1}\left(\widetilde{\mu}_4^{(1)} - 1\right)\exp\left(\frac{1}{2}\sum_{j=2}^{d-1}\frac{1}{n_j}\right).$$

Combining this with (33), (39) and setting

$$\eta := \frac{\#\left\{m_1 \neq m_2 \mid q_{m_1} = q_{m_2}\right\}}{M(M-1)} = \frac{(n_0 - 1)n_0 n_d}{n_0 n_d (n_0 n_d - 1)} = \frac{n_0 - 1}{n_0 n_d - 1},$$

we obtain

$$\mathbb{E}\left[\widehat{\mathrm{Var}}[Z^2]\right] \geq \frac{1}{n_0^2}\left(1 - \frac{1}{M}\right)\left(\eta + \frac{4\left(1 - \eta\right)}{n_1}\left(\widetilde{\mu}_4^{(1)} - 1\right)e^{-\frac{1}{n_1}}\right)\exp\left(\frac{1}{2}\sum_{j=1}^{d-1}\frac{1}{n_j}\right),$$

proving (15). Finally, the upper bound in (14) follows from dropping the negative term in (33) and applying the upper bound from (12). $\qquad\square$