[Reviews · NeurIPS 2018]

Reviewer 1



The paper studies the gradient vanishing/exploding problem (EVGP) theoretically in deep fully connected ReLU networks. As a substitute for ensuring if gradient vanishing/exploding has been avoided, the paper proposes two criteria: annealed EVGP and quenched EVGP. It is finally shown that both these criteria are met if the sum of reciprocal of layer widths of the network is a small number (thus the width of all layers should ideally be large). To confirm this empirically, the paper uses an experiment from a concurrent work. Comments: To motivate formally studying EVGP in deep networks, the authors refer to papers which suggest looking at the distribution of singular values of the input-output Jacobian. If the singular values are far from 1 and take both small and large values, it implies the network is suffering from exploding gradient or vanishing gradient problem or both. However, to study the Jacobian analytically, the authors propose to study the annealed and quenched criteria, which in some sense measure how much fluctuation the Jacobian has across its elements. The analysis suggests a high fluctuation is bad while small fluctuation avoids EVGP. If I understand the paper correctly, my main concern with the proposed criteria is that having a small fluctuation in the Jacobian across its elements would mean all elements are similar. In this case, all but one sigular values of the Jacobian would collapse to ~0, which is bad. So the proposed criteria seems problematic to me. Regarding the cited experiment for an empirical justification of the proposed theory, the paper from which the figure is taken proposes an alternative theoretical reason for this behavior. This makes it hard to understand whether a more appropriate reason for this behavior is the justification in the original paper or this paper. The paper is clearly written and original in the sense that it proposes to study EVGP but may be problematic. Comments after reading the rebuttal: I am still not sure I understand how the input-output Jacobian Z is well conditioned (singular values are close to one) if each entry of the Jacobian Z_pq^2 has a finite expected value (Theorem 1 Eq 10) and variance as shown in Eq 11 (of course these results are extended to talk about the empirical variance of the squared entries of Z in theorem 2 and 3). For instance, consider Z to be of dimension 400x800. If I generate the entries of Z by sampling from a gaussian with mean 1/400 and variance 1/400 (I have assumed n0=400 and width of the network layers to be infinite in theorem 1 to roughly use these numbers), I get a matrix Z that has singular values spanning the range 0.02 to 1.4 (thus condition number 1.4/0.02=70>>1). Another point I would like to raise is that even if the input-output Jacobian is well conditioned, it is essentially a product of terms like d h_t/d h_{t-1}, where h_t is the hidden state at depth t. Now I can have a matrix Z that is even identity, but is the result of the product of two matrices as Z = AB, where A has very large singular values and B has the inverse singular values of A, thus resulting in the singular values of 1 for Z. Here A and B signify partial derivatives like d h_t/d h_{t-1} for 2 consecutive layers. As can be seen, the gradient for layer corresponding to A are exploding while that of B will are vanishing. I feel these aspects need to be made clear in the paper.

Reviewer 2



This is nice theoretical work that establishes a relationship between the architecture of a fully connected ReLU network (namely, depth and width of its layers), and the early behavior of the gradient descent training. This early behavior in particular considers the probability of having an extreme case of exploding or vanishing gradients for random initialization of the network and shows that the variance of gradients in a fully connected ReLU net is exponential in the sum of the reciprocals of the hidden layer widths and not dependent on the biases and weights. The main strength of this work is that it provides an understanding of how architecture choices impact the training stability in non asymptotic settings (unlike previous works that discussed practically infinite layer widths). The preceding discussion suggests that to avoid the EVGP (in the annealed or quenched senses 159 described above) in deep feed-forward networks with ReLU activations. The authors state that it is best to let hidden layers grow as a function of the depth. However, two points that could be improved are the following: (1) The work right now is purely theoretical, without any empirical example or evidence. It would be good to add some example/s demonstrating the applicability of the presented theorems for tuning the architecture of a network, especially when balanced against other problem-related criteria. (2) The paper focuses on a specific, somewhat oversimplified, network architecture (i.e., feed forward, fully connected, ReLU-based networks) How do the results relate to other architectures that were proposed - the authors mention in passing that the result does not apply, for example. to convolutional or recurrent networks, but still some further discussion about this point and the impact of such design choices on the presented bounds would be useful for the readers. The paper is clear and well written.

Reviewer 3



This paper provides mathematical results regarding the link between the structure of a deep neural network and its tendency to be affected by Exploding or Vanishing Gradients. It estimates this tendency using the Jacobian matrix of the function represented by the neural network, under the rationale that the EVG problem is directly linked to the spectrum of this Jacobian and more specifically the fact that it has very large and very small singular values. The authors study the statistical distribution of the coefficients of this matrix along two axes. First (Theorem 1) by focusing on a single coefficient of the matrix and studying its second, fourth and higher moments when considered as a random variable across initialization of the neural network. Second (Theorem 2) by focusing on a given instance of initialization, and studying the variance of the square of the coefficients across the matrix. The main result of this paper is the derivation of bounds on these values, showing that in both cases they grow exponentially with a particular quantity "beta", the sum of the inverse of the widths of the layers in the network. Thus showing that given a fan-in scaled initialization, the variance of the values of the Jacobian is mostly determined by the architecture of the network. The paper also shows hints at experimental results correlating the value of "beta" (and hence the variance of the Jacobian coefficients) with the speed at which neural network starts their training. More experimental results showing the actual impact of this quantity to the training dynamics would have been a nice addition. The main limitation of this result is that it only applies to ReLU fully-connected networks, as acknowledged by the authors. This paper describes in detail its results and their interpretation and significance. This is a good paper with interesting results towards a better understanding of the training process of neural network.